# A Defect-Inspection System Constructed by Applying Autoencoder with Clustered Latent Vectors and Multi-Thresholding Classification

**Cheng-Chang Lien *** and **Yu-De Chiu**

Department of Computer Science & Information Engineering, Chung Hua University, Hsinchu 30012, Taiwan; qq1234500@gmail.com
*** Correspondence: cclien@chu.edu.tw; Tel.: +886-3-5186404

**Abstract:** Defect inspection is an important issue in the field of industrial automation. In general, defect-inspection methods can be categorized into supervised and unsupervised methods. When supervised learning is applied to defect inspection, the large variation of defect patterns can make the data coverage incomplete for model training, which can introduce the problem of low detection accuracy. Therefore, this paper focuses on the construction of a defect-inspection system with an unsupervised learning model. Furthermore, few studies have focused on the analysis between the reconstruction error on the normal areas and the repair effect on the defective areas for unsupervised defect-inspection systems. Hence, this paper addresses this important issue. There are four main contributions to this paper. First, we compare the effects of SSIM (Structural Similarity Index Measure) and MSE (Mean Square Error) functions on the reconstruction error. Second, various kinds of Autoencoders are constructed by referring to the Inception architecture in GoogleNet and DEC (Deep Embedded Clustering) module. Third, two-stage model training is proposed to train the Autoencoder models. In the first stage, the Autoencoder models are trained to have basic image-reconstruction capabilities for the normal areas. In the second stage, the DEC algorithm is added to the training of the Autoencoder model to further strengthen feature discrimination and then increase the capability to repair defective areas. Fourth, the multi-thresholding image segmentation method is applied to improve the classification accuracy of normal and defect images. In this study, we focus on the defect inspection on the texture patterns. Therefore, we select the nanofiber image database and carpet and grid images in the MVTec database to conduct experiments. The experimental results show that the accuracy of classifying normal and defect patch nanofiber images is about 86% and the classification accuracy can approach 89% and 98% for carpet and grid datasets in the MVTec database, respectively. It is obvious that our proposed defect-inspection and classification system outperforms the methods in MVTec.

**Keywords:** defect inspection; autoencoder; SSIM; DEC clustering algorithm

## 1. Introduction

Defect inspection is an indispensable process in the field of industrial automation. To identify these defect regions, the defect area, color variation, and texture complexity are important factors that can affect the accuracy of defect inspection. In recent years, due to the development of deep-learning technology, many deep-learning models for defect inspection have been proposed, mainly divided into supervised learning and unsupervised learning. Supervised defect-inspection methods can be categorized into bounding-box-based methods and pixel-based methods. The representative bounding-box-based methods are the YOLO series [1–4]. For example, research in [5–9] used different versions of YOLO architectures to detect the defect regions and their experiments show that both of the classification and region locating are accurate. For the pixel-based method, SegNet [10]

can segment the defect area precisely and distinguish the defect category. However, the computing efficiency is low. Mask R-CNN [11] technology combines both bounding-box-based and pixel-based methods. First, the defect region is located with the bounding box, and then the precise defect region is segmented with high computational efficiency.

However, training a supervised model requires a large number of labeled defect images, and the data pre-processing will consume a lot of time and human resources. In addition, the collection of defect images faces two major difficulties [12–14]. First, it is difficult to collect enough defect patterns to have high data coverage for model training; second, because the occurrence of defects is unpredictable, it is difficult to determine whether the collected samples cover all defect patterns. These problems can cause missing detection or poor accuracy in defect inspection. Therefore, this paper focuses on unsupervised learning.

Most of the unsupervised learning techniques use the Autoencoder architecture [15] shown in Figure 1 to learn from normal images. The general Autoencoder model only has the capability to reconstruct normal images. However, it can only slightly repair the texture or color on the defect region shown in Figure 2. Therefore, in addition to training the Autoencoder model, it is also necessary to perform model training through methods such as clustering of feature vectors (latent vectors), to improve the model's capability to repair the defect area.

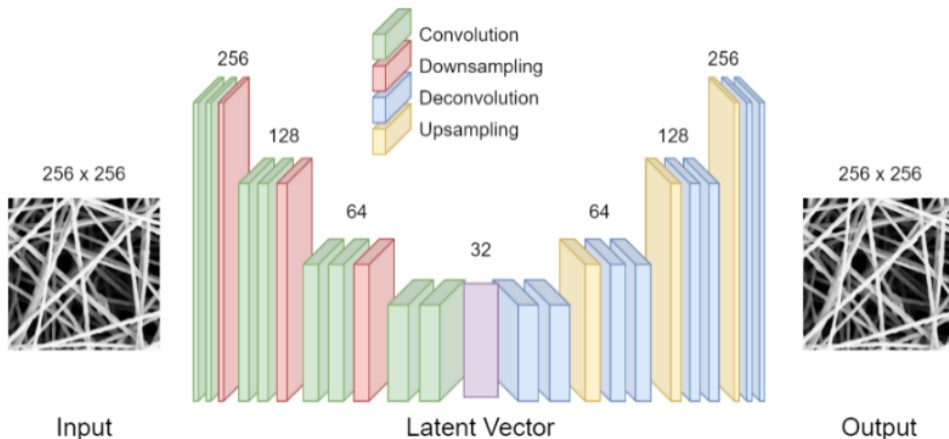

**Figure 1.** Structure of Autoencoder.

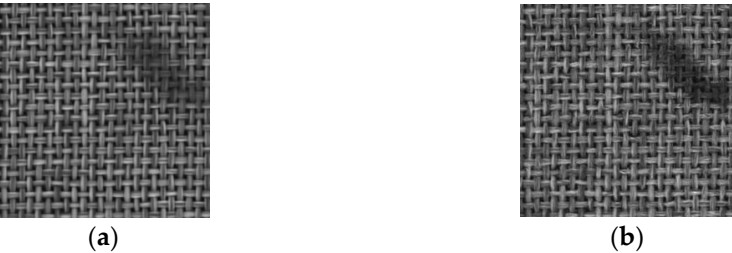

| (a) | (b) |

**Figure 2.** (**a**) A test defect image. (**b**) Reconstructed image through general Autoencoder.

For the related methods that combine Autoencoder and enhanced defect-detection technologies, the works in [16,17] train the VAE model [18,19] with the KL-Divergence function [20] to make the distribution of latent vectors approach Gaussian distribution. Therefore, when the test image contains a defect region, the defect region can be repaired with a correct texture pattern. However, it is found in experiments that it is difficult to balance the Gaussian distribution parameters between the normal area reconstruction and the defect area repair. As an extension of VAE technology [21], GMVAE [22] technology produces a more obvious clustering effect on Latent vectors.

Literature [12] proposed a defect image classification method by randomly selecting some images from the training dataset and generating feature vectors from the encoder as

the reference samples. The feature vectors of the test images are generated in the same way. The feature vectors of the reference sample and the test image are compared to confirm whether the test image is a defect image. This method simply judges whether the test image is a defect image, and does not discuss how to detect the defect region. The MS-FCAE [13] technology uses a multi-scale architecture to simultaneously extract features across the multi-scale feature maps. Furthermore, on each scale, the clustering algorithm [23] is applied to train the model to have a better defect repair capability.

Most of the above papers focused on the detection and repair capabilities of defect areas, and seldom discussed how to classify normal and defect images. MVTec [14] built a database for defect inspection, including 5 types of texture images and 10 object images, and evaluated the detection capabilities of 6 defect detection models for 15 types of images. This paper proposes a classification method for normal and defect images, but its accuracy is low. Therefore, in addition to improving the accuracy of repairing defect areas, this paper also focuses on proposing effective methods to improve the accuracy of classification of normal and defect images. Here, we use carpet and grid datasets in the MVTec database to conduct experiments, and compare the performance between the proposed method in this paper and the four models in MVTec [14].

As the Inception network architecture of GoogleNet [24–27] can provide multi-scale feature-extraction capabilities, we refer to the Inception network to design a new Autoencoder model to improve the feature-extraction and texture-reconstruction capabilities. For normal image reconstruction, we apply SSIM (Structural Similarity Index Measure) [28] as the loss function for network training. For the repair of the defect region, we also use the KL-Divergence function [13,23] for clustering Latent feature vectors, and enlarge the distance among different clusters, so as to obtain better defect area repair capability. The Inception Autoencoder and traditional Autoencoder networks proposed in this paper will be trained in a two-stage manner. In the first stage, the Autoencoder model is trained with normal images to obtain the basic image-reconstruction capability, and in the second stage, the Autoencoder model combined with a clustering module is trained to obtain the defect area repair capability. Hence, this paper has the following main contributions:

1. We construct four kinds of Autoencoder models by referring to the structures of Inception architecture in GoogleNet [24–27] and combine the DEC clustering technology [13,23] to strengthen the discrimination of the latent vectors. Then, the defect-detection performance including the normal-area-reconstruction error and defect area repair capability are analyzed. Furthermore, SSIM is applied to calculate the similarity between the reconstructed image and the input image for acquiring the defect region.

2. An effective two-stage model training method is proposed to train the model with high normal-area-reconstruction accuracy and defect-area repair capability. In the first stage, the Autoencoder model is trained with a basic defect-reconstruction capability. In the second stage, the DEC loss function is combined with the Autoencoder for the clustering of latent vectors to further strengthen the feature discrimination and increase the capability of repairing defect regions.

3. A novel defect-image-classification method between normal and defective images is proposed by using the multi-thresholding image segmentation algorithm, which outperforms the method mentioned in [14]. Furthermore, with the proposed method, the defect region can be identified precisely.

The defect-image-inspection architecture is shown in Figure 3, which includes an encoder module (feature extraction), a decoder module (image reconstruction), and a feature-clustering module. The SSIM loss function is used to train both the encoder and decoder, and is also used in the testing process for computing the difference between the input and output images. Finally, the multi-thresholding module is proposed to classify normal and defect images.

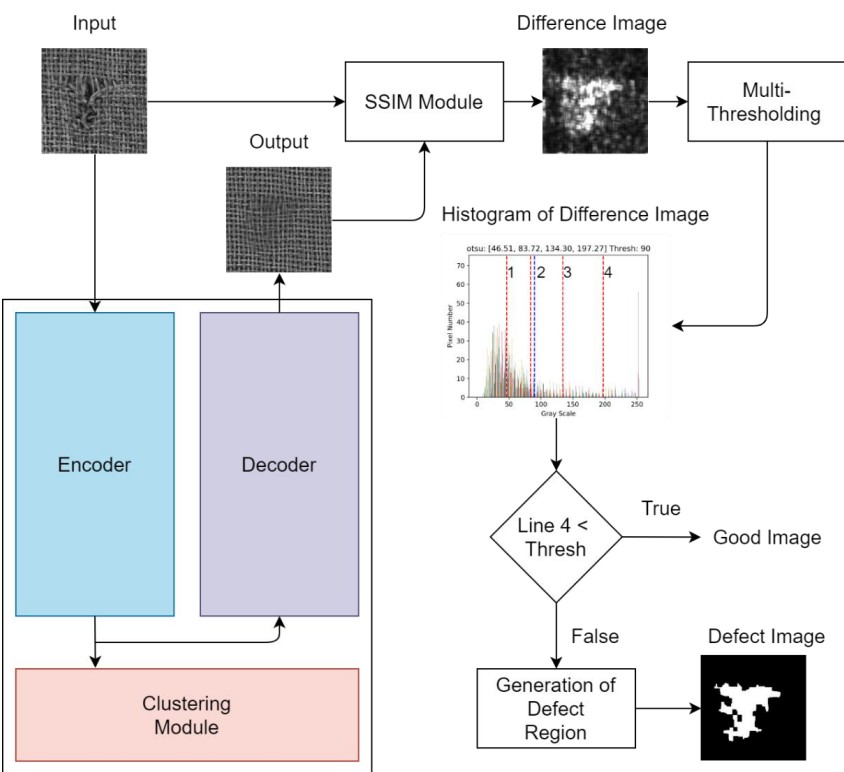

**Figure 3.** Defect image detection architecture. The input defect image is reconstructed by the proposed Autoencoder, and the defect area is repaired with the correct texture pattern. The SSIM matching function is used to compute the difference between the input and output images, and then the multi-threshold image segmentation algorithm is used to classify the normal and defect images.

## 2. Autoencoder for Normal Image Reconstruction

In order to improve the reconstruction accuracy of normal images, we proposed the Autoencoder architecture with two different kinds of key modules as the unsupervised learning model, as shown in Figure 4. In the following, we will introduce the proposed Autoencoder in detail.

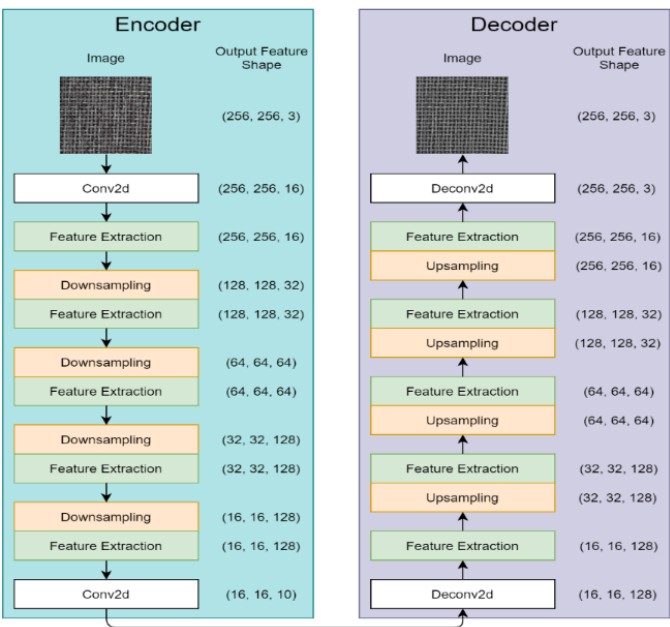

**Figure 4.** Structure of the proposed Autoencoder model.

### 2.1. Structure of Encoder

The encoder architecture designed in this paper is shown in the encoder part of Figure 4. The encoder extracts and encodes image features in the latent space. The depth of the model will affect whether the learned feature maps are discriminative. However, a model that is too deep will make it difficult to retain image details, resulting in blurred reconstructed images, and even overfitting. Therefore, the design of the model must strike a balance between depth and detail preservation.

The encoder first applies a $1 \times 1$ filter to the input image to increase the depth from 3 to 16 to increase the feature capacity in the first layer. Then, downsampling and feature extraction submodules are combined to form a basic module in the encoder, and this module is repeatedly used to generate the Latent feature vectors. In addition, in the downsampling process (dimensionality reduction), while the length and width of the feature map are reduced by half, the depth is doubled to avoid excessive feature-loss to maintain the image detail. Finally, the $1 \times 1$ Conv2d module is used to generate latent vectors.

The two submodules, feature extraction and downsampling are the core of the encoder, and their architecture can be modified according to the image complexity. In this study, we design the simple convolution module and Inception convolution module as the feature extraction modules with different complexities. The feature-extraction module can be constructed with a simple convolution module, shown in Figure 5a, by using only one layer of the Conv2d module, or refer to the architecture of Inception v4 convolution module [27], shown in Figure 5b. Compared with the simple convolution module, the Inception convolution module is composed of four different branches designed to capture features across different scales.

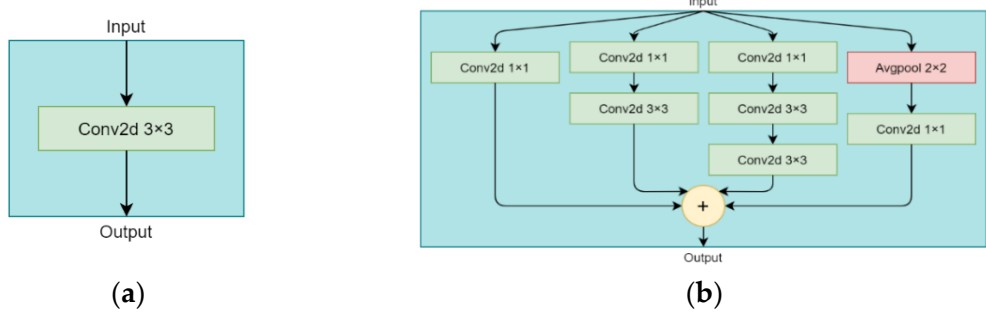

(**a**)  (**b**)

**Figure 5.** (**a**) Simple convolution module. (**b**) Inception convolution module.

The main function of the downsampling module is to decimate the size of the feature map and increase the depth to twice the original. Figure 6a shows a downsampling module with a relatively simple structure. For reasons of preserving image details, the Inception downsampling module shown in Figure 6b uses three branches to reduce dimensionality in different ways, and finally, outputs of summation of the feature maps.

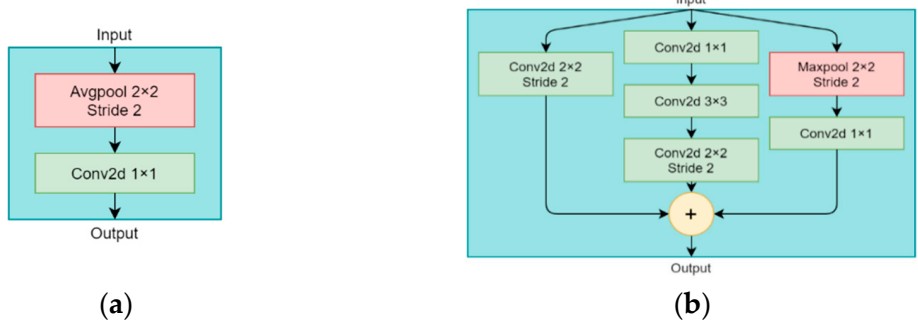

(**a**)  (**b**)

**Figure 6.** (**a**) Simple downsampling module. (**b**) Inception downsampling module.

The Conv2d module used in the above-mentioned feature extraction and the down-sampling modules is designed as the structure shown in Figure 7, which includes the convolution, batch normalization, and Relu activation operations.

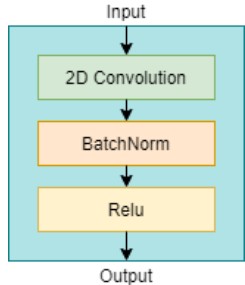

**Figure 7.** Structure of Conv2d module.

### 2.2. Structure of Decoder

The architecture of the decoder is shown on the right part of Figure 4. Its structure is a mirror version of the encoder, and its function is to reconstruct the input image from the Latent feature map ($16 \times 16 \times 10$). First, the depth of the feature map is increased from 10 to 128 through a Deconv2d module and then the deconvolution and upsampling modules shown in Figures 8 and 9, respectively, are used to reconstruct the input image iteratively. While reconstructing defect image, the length and width are gradually doubled in size, and the depth is gradually reduced by half. Finally, the Deconv2d module is used to generate the output image. The Deconv2d module used in the above-mentioned modules is shown in Figure 10. This module operates in the order of deconvolution, batch normalization, and Relu activation.

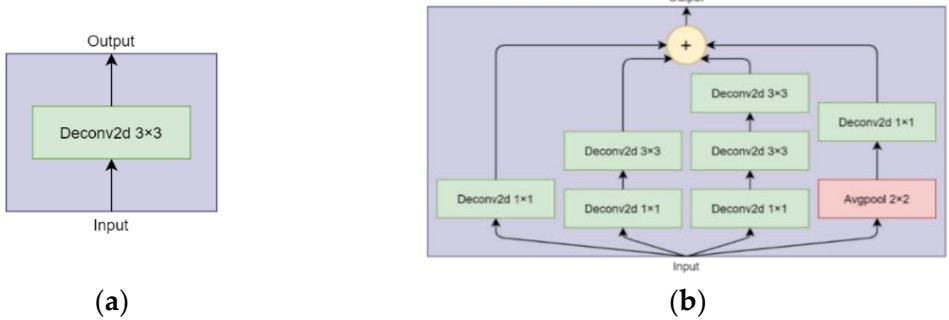

(**a**)           (**b**)

**Figure 8.** (**a**) Simple deconvolution module. (**b**) Inception deconvolution module.

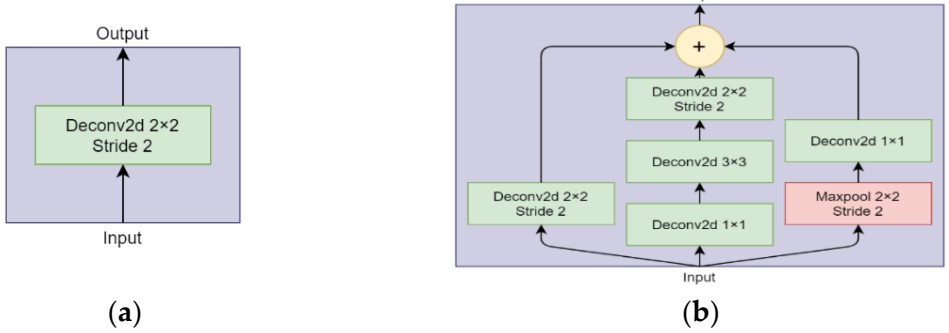

(**a**)           (**b**)

**Figure 9.** (**a**) Simple upsampling module. (**b**) Inception upsampling module.

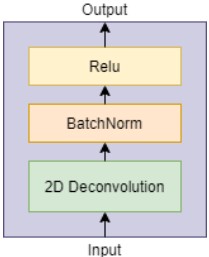

**Figure 10.** Structure of Deconv2d module.

### 2.3. Model Training for Normal Image Reconstruction

In this study, first, we train Autoencoder with normal images to have a basic image-reconstruction capability. Instead of the MSE function, the SSIM (Structural Similarity Index Measure) function is used as the loss function for the model training. The reason is that the MSE loss function belongs to the global analysis, and the local difference information is not considered. In order to obtain the local difference between the reconstructed image and the input image, SSIM [28] is chosen as the loss function to train the model. Here, we apply the Adam [29] algorithm to update the model parameters. SSIM is defined as follows:

$$\text{SSIM}(X, Y) = l(X, Y)^\alpha c(X, Y)^\beta s(X, Y)^\gamma \tag{1}$$

where $l(X, Y)$, $c(X, Y)$, $s(X, Y)$ represent luminance, contrast, and structure similarity measures, respectively, and $\alpha$, $\beta$, and $\gamma$ are the adjustment parameters. The three similarity measure functions are defined as follows:

$$l(X, Y) = \frac{2\mu_X\mu_Y + C_1}{\mu_X^2 + \mu_Y^2 + C_1}, c(X, Y) = \frac{2\sigma_X\sigma_Y + C_2}{\sigma_X^2 + \sigma_Y^2 + C_2}, s(X, Y) = \frac{2\sigma_{XY} + C_3}{\sigma_X\sigma_Y + C_3}. \tag{2}$$

where $\mu_X$ and $\mu_Y$ are the average values of the images $X$ and $Y$, $\sigma_X$ and $\sigma_Y$ are the standard deviations of the images $X$ and $Y$, and $\sigma_{XY}$ is the cross covariance of the two images. $C_1$, $C_2$, and $C_3$ are all constants, which are used to adjust the weight of the three items. To simplify Equation (2), the parameters are set as $\alpha = \beta = \gamma = 1$ and $C_3 = C_2/2$, and then the following simplified version can be obtained:

$$L_{\text{SSIM}} = \text{SSIM}(X, Y) = \frac{(2\mu_X\mu_Y + C_1)(2\sigma_{XY} + C_2)}{(\mu_X^2 + \mu_Y^2 + C_1)(\sigma_X^2 + \sigma_Y^2 + C_2)}. \tag{3}$$

However, only using SSIM loss function to train the Autoencoder model without the Latent vector clustering can have a good reconstruction capability for normal images, but has a low repair capability for defective images.

### 3. Autoencoder Model for Repairing Defect Areas

The Autoencoder for repairing the defect area is established based on the clustering of latent vectors. When a defect image is input to the model, the latent vectors within the defect area can be corrected toward the specific cluster center of the normal image, thereby repairing the defect area and highlighting the defect area. However, the optimization of defect area repair (the degree of feature grouping) and normal image-reconstruction quality are contradictory to each other. Therefore, the optimization of the two must be a trade-off in order to achieve the best effect in defect inspection.

### 3.1. Autoencoder with Latent Vector Clustering

The Autoencoder model with a defective area repair capability is shown in Figure 11. Three modules will be integrated, namely the encoder module, decoder module, and latent vector clustering module. The encoder and decoder also follow the structure mentioned

in the previous section, and the clustering module establishes the capability to repair the defective area.

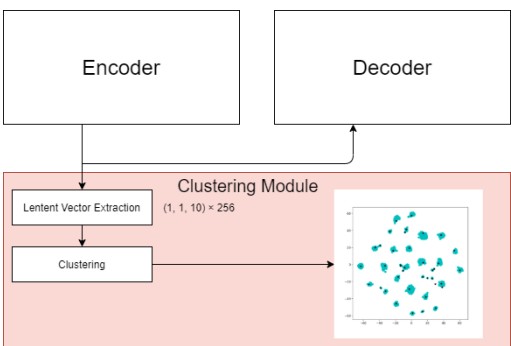

**Figure 11.** The Autoencoder model with defect area repair capability.

Before using the clustering module to start training the model, the clustering module needs to initialize the cluster centers with the K-means clustering algorithm. First, the feature map ($16 \times 16 \times 10$) is segmented into 256 latent vectors with dimension $1 \times 1 \times 10$. However, the dimension of the latent vector is not necessarily limited to the size of $1 \times 1 \times 10$. The dimension of the latent vector depends on the texture complexity of the input image. In this paper, we randomly select 100,000 feature vectors to initialize the K-means clustering algorithm, and the number of clusters is preset to 40 clusters.

### 3.2. Deep Embedded Clustering Algorithm

The purpose of feature clustering training using the DEC algorithm [23] is to concentrate the probability distribution of latent vectors from scattered to concentrated probability distribution. The DEC algorithm uses the distance between the latent vector and the cluster center to calculate the probability that the latent vector belongs to each cluster, and it is called the source probability distribution. The source probability that the *i*-th latent vector belongs to the *j*-th cluster is defined as Equation (4).

$$S_{ij} = \frac{\left(1 + ||c_i - \mu_j||^2/\alpha\right)^{-\frac{\alpha+1}{2}}}{\sum_{j'}\left(1 + ||c_i - \mu_{j'}||^2/\alpha\right)^{-\frac{\alpha+1}{2}}} \tag{4}$$

where $c_i$ is the *i*-th latent vector and $\mu_j$ is the *j*-th cluster center. The value of $\alpha$ is used to adjust the clustering degree of the source probability distribution $S_{ij}$. The target probability distribution defines the target that the source probability distribution approaches, as in Equation (5).

$$T_{ij} = \frac{S_{ij}^2/f_j}{\sum_{j'} S_{ij'}^2/f_{j'}}, \ f_j = \sum_i S_{ij} \tag{5}$$

Finally, the KL-Divergence function defined in Equation (6) is used to describe the difference between $S_{ij}$ and $T_{ij}$ as the loss function.

$$L_{DEC} = \text{KL}(\text{T}||\text{S}) = \sum_i \sum_j T_{ij} log \frac{T_{ij}}{S_{ij}} \tag{6}$$

Figure 12 illustrates the clustering effect for the latent vectors generated from the encoder trained with the KL-Divergence function.

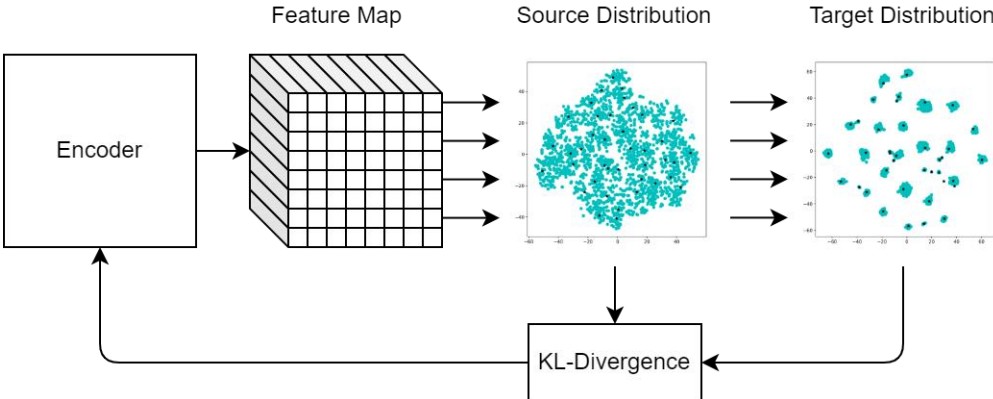

**Figure 12.** The KL-Divergence function is used to modify the parameters of the encoder to cluster the latent vectors.

In order to make the Autoencoder model have both normal area reconstruction capability and defect area repair capability, this paper proposes a two-stage training method. The first stage uses the SSIM loss function to train the encoder and decoder modules to make them have normal area reconstruction capabilities. In the second stage of model training, different combinations of loss functions are used to optimize each module to make it have the capability to repair defective areas. The procedures of the model training method are described in the following steps:

Step 1      Train Encoder Module + Decoder Module with $L_{\text{SSIM}}$
Step 2.1    Train Encoder Module with $\alpha_{\text{SSIM}}\, L_{\text{SSIM}} + L_{\text{DEC}}$
Step 2.2    Train Decoder Module with $L_{\text{SSIM}}$
Step 2.3    Train Clustering Module with $L_{\text{DEC}}$

Since the encoder module needs to output the feature map to the decoder module and the clustering module, it is necessary to use $L_{\text{SSIM}}$ and $L_{\text{DEC}}$ functions at the same time, and use $\alpha_{\text{SSIM}}$ to adjust the ratio between the two functions to optimize the training results. The decoder module is only responsible for reconstructing the latent map back to the image, so it only needs $L_{\text{SSIM}}$ function to train the decoder. Finally, the clustering module is trained with $L_{\text{DEC}}$ function to update the cluster centers and KL-Divergence function.

### 3.3. The Classification of Normal and Defect Images

The flowchart of defect-image detection is shown in Figure 3. When a defect image is input to the modified Autoencoder, the system will reconstruct the normal area and repair the defect area with the texture of the normal image. Then, we use SSIM to calculate the similarity between the input image and the output image to generate the difference image. As there are still reconstruction errors in the normal and defect repairing areas within the difference image, the pixel value distribution of the difference image is variated. Therefore, it is very difficult to determine a fixed threshold to determine whether the input image is normal or defective. In this study, we try a variety of methods to identify defect images, and the analyses show that the multi-thresholding method can have the highest accuracy. Here, we apply the Otsu [30,31] multi-thresholding method to calculate four thresholds, and the position of the fourth threshold (the fourth red dotted line in the difference grayscale distribution in Figure 3) is used to determine whether the input image is a normal or defect image. If the position of the fourth threshold is larger than the specified threshold value, then this image is classified into a defect image. If the input image is classified as a defect image, the segmented areas on the difference image with third and fourth thresholds will be fused together to determine the defect region shown in Figure 13.

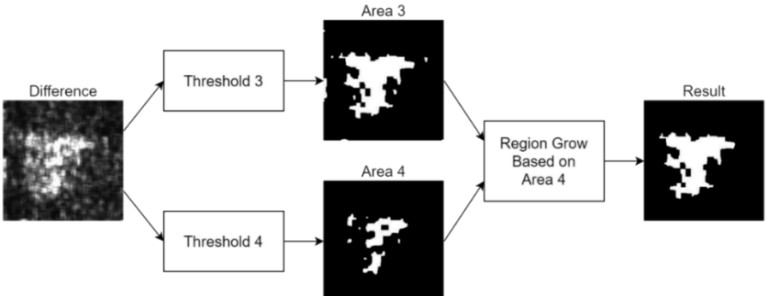

**Figure 13.** If the input image is classified as a defect image, the third and fourth thresholds will be used to binarize the difference image to mark the defect area.

## 4. Experimental Results

In this study, we focus on the defect inspection of texture patterns. Therefore, we select the Nanofiber image database [32] and carpet and grid images in the MVTec database [14] to conduct experiments. The nanofiber image database provides fewer normal images for training and testing. Therefore, data augmentation is needed to increase the normal images. The MVTec database contains mainly 5 types of texture images and 10 types of object images. In this study, we select two types of textures: carpet and grid to conduct experiments. The details of these databases are described in Tables 1–3. In this section, loss function analysis for normal area reconstruction and defective area repair, weighting analysis between SSIM and DEC loss functions, accuracy analysis for classifying normal and defect test images, subjective evaluation of defective area detection, and the comparison between our proposed method and the SOTA methods in [14] are conducted to verify the performance of the proposed method. Our system uses an NVIDIA TITAN Xp GPU, and the computing efficiency is about 32 FPS.

**Table 1.** Dataset of nanofiber image.

|  | Training Data | Testing Data | |
|---|---|---|---|
| *Category* | Good | Good | Defect |
| *Amount (frame)* | 4 | 1 | 40 |
| *Image size* | $1024 \times 700$ | $1024 \times 700$ | $1024 \times 696$ |

**Table 2.** Dataset of carpet image.

|  | Training Data | Testing Data | | | | |
|---|---|---|---|---|---|---|
| *Category* | Good | Good | Color | Cut | Hole | Metal | Thread |
| *Amount (frame)* | 277 | 28 | 19 | 17 | 17 | 17 | 19 |
| *Image size* | | | $1024 \times 1024$ | | | | |

**Table 3.** Dataset of grid image.

|  | Training Data | Testing Data | | | | |
|---|---|---|---|---|---|---|
| *Category* | Good | Good | Bent | Broken | Glue | Metal | Thread |
| *Amount (frame)* | 264 | 21 | 12 | 12 | 11 | 11 | 11 |
| *Image size* | | | $1024 \times 1024$ | | | | |

*4.1. Loss Function Analysis for Normal Area Reconstruction and Defective Area Repair*

In this experiment, the training effects of SSIM and MSE loss functions are compared, and experiments are conducted on both the Simple Autoencoder and the Inception Autoencoder. Furthermore, SSIM and MSE are also used as the evaluation functions to compare

their accuracy for reconstructing normal areas and repairing defective areas. Tables 4 and 5 are the simulation results of Simple and Inception Autoencoders for normal area reconstruction. The experimental results show that whether using SSIM or MSE loss function for training, in most cases, as long as the same function is used for testing, there will be good performance. In addition, the data in Tables 4 and 5 show that the Inception Autoencoder has better reconstruction accuracy on normal area, that is, it has a better capability to preserve detailed textures.

**Table 4.** Analysis of reconstruction error using Simple Autoencoder for normal image.

| | Nanofiber | | Carpet | | Grid | |
|---|---|---|---|---|---|---|
| | Testing with SSIM Matching Function | Testing with MSE Matching Function | Testing with SSIM Matching Function | Testing with MSE Matching Function | Testing with SSIM Matching Function | Testing with MSE Matching Function |
| *Training with SSIM loss function* | 0.1440 | 479.05 | 0.1171 | 202.09 | 0.0361 | 21.50 |
| *Training with MSE loss function* | 0.1739 | 458.81 | 0.1386 | 177.62 | 0.0554 | 29.77 |

**Table 5.** Analysis of reconstruction error using Inception Autoencoder for normal image.

| | Nanofiber | | Carpet | | Grid | |
|---|---|---|---|---|---|---|
| | Testing with SSIM Matching Function | Testing with MSE Matching Function | Testing with SSIM Matching Function | Testing with MSE Matching Function | Testing with SSIM Matching Function | Testing with MSE Matching Function |
| *Training with SSIM loss function* | 0.1287 | 403.63 | 0.1055 | 178.77 | 0.0322 | 19.92 |
| *Training with MSE loss function* | 0.1400 | 348.88 | 0.1277 | 168.04 | 0.0455 | 23.70 |

Figures 14–16 show the subjective evaluation of the reconstruction quality using SSIM and MSE as the training and testing functions on the normal images. In Figure 14, it can be clearly seen that the reconstruction quality of SSIM is better than that of MSE for nanofiber images. In Figures 15 and 16, the texture of the carpet and grid images is relatively simple, and the reconstruction error is small. Therefore, there is no significant difference between SSIM and MSE in the test.

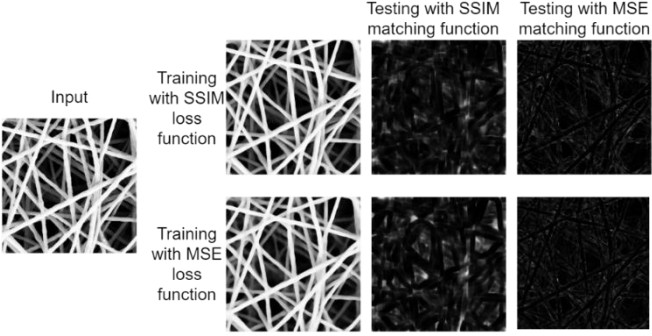

**Figure 14.** The subjective evaluation of the reconstruction quality using SSIM and MSE as the training and testing functions on the normal nanofiber images.

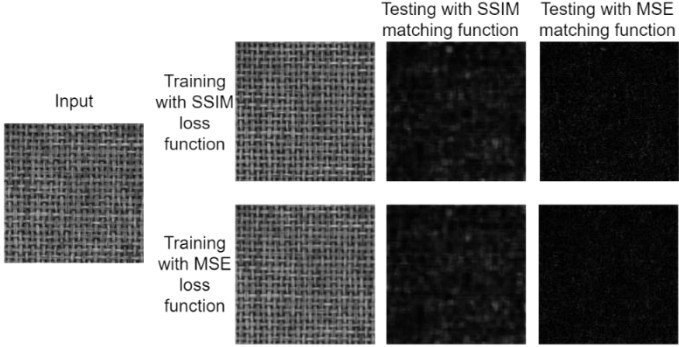

**Figure 15.** The subjective evaluation of the reconstruction quality using SSIM and MSE as the training and testing functions on the normal carpet images.

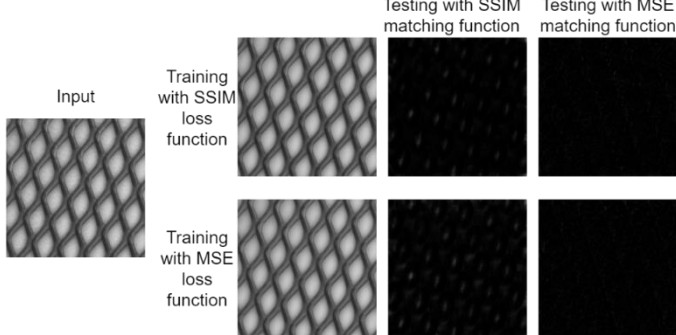

**Figure 16.** The subjective evaluation of the reconstruction quality using SSIM and MSE as the training and testing functions on the normal grid images.

Figures 17–19 show the subjective evaluation of defect-repair quality using SSIM and MSE as the training and testing functions on defect images. Figure 17 shows that using SSIM is better for texture repair in the defective area for nanofiber images. It is also obvious that the defective area can be clearly found using SSIM, but the defective area cannot be found using MSE. Figure 18 shows that selecting SSIM as the loss function has a better repair effect on the defective area on the carpet images, and furthermore, can mark the defect position more completely on the difference image. In Figure 19, the simulation results show that MSE and SSIM can provide good repair quality for the grid dataset. It can be observed from Figures 17–19 that the complexity of the image texture will affect the effect of repairing the defective area. The more complex the image texture, the worse the repairing effect of the defect area, as can be observed in Figure 17. The simpler the image texture, the better the repairing effect of the defect area, which can be observed in Figures 18 and 19. In general, compared with MSE, SSIM is more accurate in detecting the defective area.

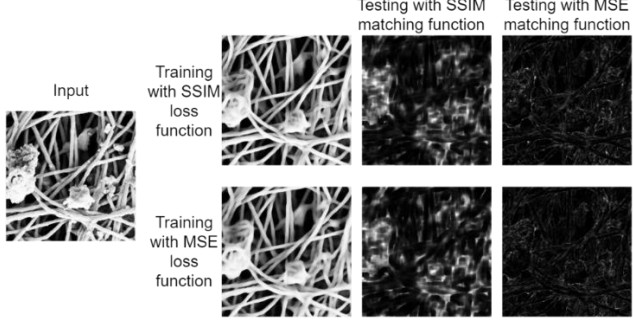

**Figure 17.** The subjective evaluation of repair quality using SSIM and MSE as the training and testing functions on defective nanofiber images.

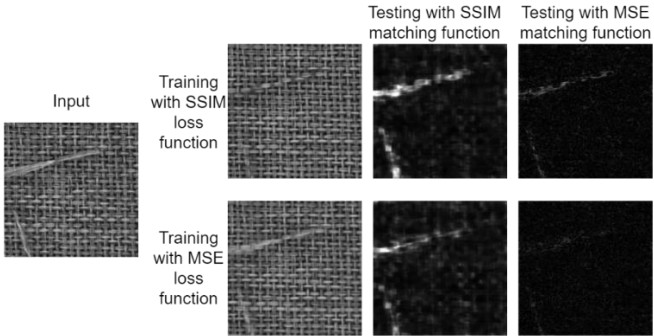

**Figure 18.** The subjective evaluation of repair quality using SSIM and MSE as the training and testing functions on defective carpet images.

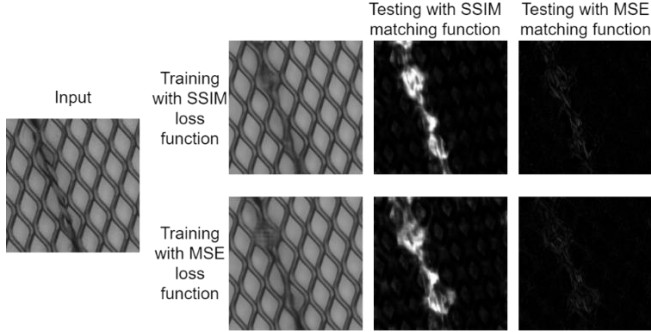

**Figure 19.** The subjective evaluation of repair quality using SSIM and MSE as the training and testing functions on defective grid images.

### 4.2. Weighting Analysis between SSIM and DEC Loss Functions

The SSIM loss function affects the reconstruction similarity on the normal area and the DEC loss function affects the repair effect on the defective area. The optimization directions of the two for the model training are opposite. A higher weight of SSIM loss function improves the reconstruction similarity on the normal area, but it will cause the defect area to be unable to be repaired. This phenomenon is shown in Figure 20. Since the defect area cannot be completely repaired back to the normal image as shown in Figure 20b, the difference in the defective area is not obvious in the difference image shown in Figure 20c, and then it will be judged as a normal image. Increasing the weight of the DEC loss function improves the repair capability of the defective area but reduces the reconstruction effect of the normal area, and many noisy regions, as shown in Figure 21, appear. It can be seen that a balance needs to be struck between the weights of SSIM and DEC loss functions.

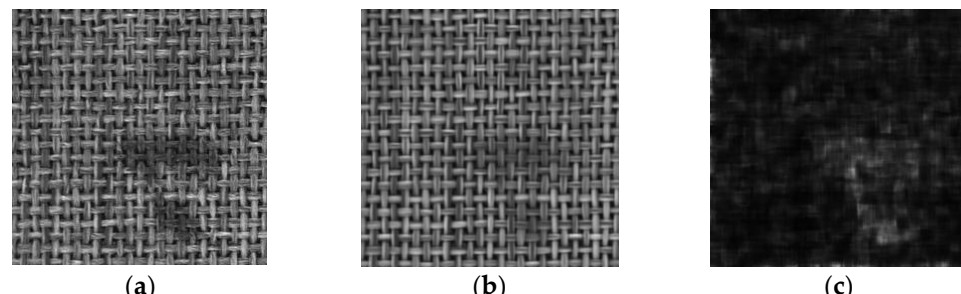

| (a) | (b) | (c) |

**Figure 20.** Simulation result for higher weight of SSIM loss function. (**a**) The defective carpet image. (**b**) The reconstructed image for image (**a**). (**c**) The difference image between images (**a**,**b**) obtained by SSIM function.

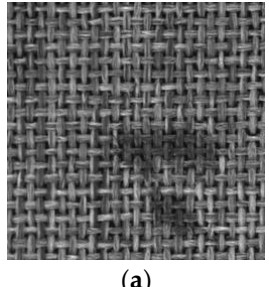 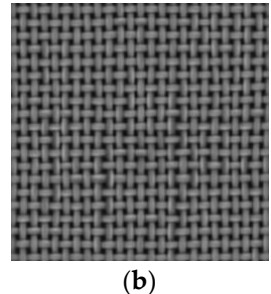 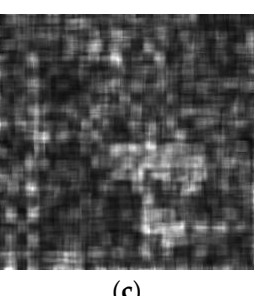

| (a) | (b) | (c) |

**Figure 21.** Simulation result for higher weight of DEC loss function. (**a**) The defective carpet image. (**b**) The reconstructed image for image (**a**). (**c**) The difference image between images (**a**,**b**) obtained by SSIM function.

In Section 2, we show that the Inception Autoencoder has a much higher accuracy in normal area reconstruction than the Simple Autoencoder. This also means that when the Inception Autoencoder is used as a defect-inspection model, it must be assigned a lower weight of SSIM loss function than the Simple Autoencoder to achieve better normal area reconstruction and defect area repair effects. In Table 6, we suggest weighting ratios between SSIM and DEC loss functions for Simple Autoencoder and Inception Autoencoder.

**Table 6.** Weighting ratios between SSIM and DEC loss functions for Simple and Inception Autoencoders.

|  | **Nanofiber** | *Carpet* | *Grid* |
| --- | --- | --- | --- |
| *Model* | | SSIM:DEC | |
| *Simple AE* | 300:1 | 140:1 | 80:1 |
| *Inception AE* | 110:1 | 80:1 | 100:1 |

In addition, in the DEC method, the number of clusters has a significant impact on the repairing effect on the defective area. In Figure 22, we illustrate the clustering effect for different cluster numbers by using the latent vectors of carpet images. Figure 22a shows that some large clusters can be further grouped into more clusters. However, Figure 22c shows there are many empty clusters when the cluster number is too large. Hence, we set the number of clusters $k$ to 40 that can have a better repairing effect on the defective area.

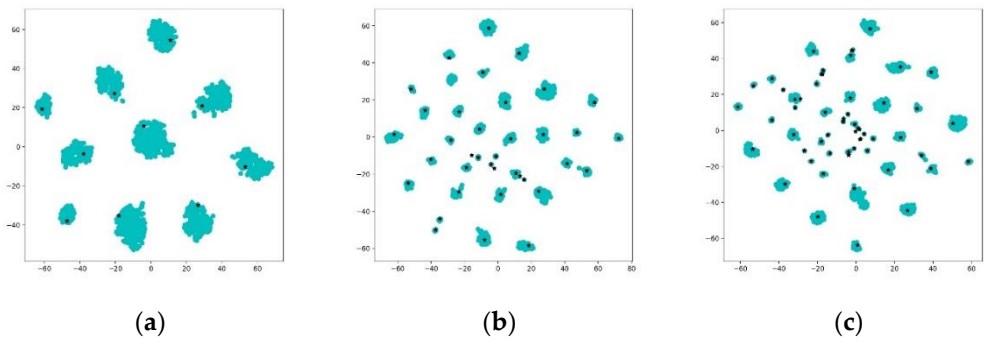

| (a) | (b) | (c) |

**Figure 22.** The clustering analysis for different cluster number by using the latent vectors of carpet images. (**a**) k = 10. (**b**) k = 40. (**c**) k = 60.

### 4.3. Accuracy Analysis for Classifying Normal and Defect Images

Most defect-inspection systems [13,16,17,22] are focused on the detection and repair capabilities of defect areas, and it is seldom discussed how to classify normal and defect images under image-reconstruction errors. Here, we use three kinds of datasets (nanofiber, carpet, and grid) to conduct the experiments of classifying normal and defect patch images. The Autoencoder models used in this experiment include Simple Autoencoder (SAE),

Simple Autoencoder + DEC (SAE + DEC), Inception Autoencoder (IAE), and Inception Autoencoder + DEC (IAE + DEC). Since the proposed Autoencoder models use patch images for model training and testing, the classification accuracy analysis of normal and defective images comprises patch image analysis and complete image analysis. The classification method for normal and defective images is described in Section 3.3, and the classification thresholds for three datasets are listed in Table 7.

**Table 7.** Classification thresholds for three datasets.

|  | **Nanofiber** | *Carpet* | *Grid* |
|---|---|---|---|
| *Threshold* | 90 | 90 | 80 |

Tables 8–10 show the classification accuracy of each kind of dataset. The classification method is determined by whether the fourth Otsu threshold exceeds the classification threshold. If the threshold is greater than the classification threshold, it is a defective image, and if it is less than the classification threshold, it is a normal image. The accuracy of classifying good and defect images is similar to that of MVtec [14], which is defined as the number of correctly classified test images divided by the number of test images. Observing Tables 8–10, it can be found that SAE and IAE have better normal image classification accuracy when the DEC loss function is not added, but the classification of defective images is worse. After adding the DEC loss function, SAE and IAE can improve the accuracy of normal and defective image identification, and the accuracy of defective image classification can be significantly improved. However, on more complex nanofiber images, the classification accuracy of normal images is reduced because the DEC loss function can reduce the reconstruction capability on normal areas, as listed in Table 8. It is interesting that SAE + DEC outperforms IAE + DEC on complex nanofiber images.

**Table 8.** Accuracy analysis of classifying normal and defect patch nanofiber images.

| *Model* | **Good** | **Defect** |
|---|---|---|
| *SAE* | 90.10% | 65.13% |
| *SAE + DEC* | 82.41% | 90.79% |
| *IAE* | 100% | 67.31% |
| *IAE + DEC* | 75.82% | 86.68% |

**Table 9.** Accuracy analysis of classifying normal and defect patch carpet images.

| *Model* | **Good** | **Color** | **Cut** | **Hole** | **Metal** | **Thread** |
|---|---|---|---|---|---|---|
| *SAE* | 100% | 40.57% | 92.50% | 90.14% | 93.54% | 90.69% |
| *SAE + DEC* | 99.19% | 100% | 100% | 100% | 100% | 100% |
| *IAE* | 99.19% | 21.73% | 88.75% | 83.09% | 91.93% | 79.06% |
| *IAE + DEC* | 99.59% | 91.30% | 100% | 98.59% | 100% | 98.83% |

**Table 10.** Accuracy analysis of classifying normal and defect patch grid images.

| *Model* | **Good** | **Bent** | **Broken** | **Glue** | **Metal** | **Thread** |
|---|---|---|---|---|---|---|
| *SAE* | 98.94% | 88.23% | 93.75% | 82.75% | 100% | 98.57% |
| *SAE + DEC* | 100% | 100% | 100% | 100% | 100% | 100% |
| *IAE* | 100% | 85.29% | 81.25% | 58.62% | 94.28% | 91.42% |
| *IAE + DEC* | 100% | 100% | 100% | 96.55% | 94.28% | 100% |

In addition, we use the AUC (area under the ROC curve) to evaluate the classification performance of the four models. Figure 23 shows the ROC curves for nanofiber, carpet, and grid datasets computed by using different classification thresholds to compare with the fourth Otsu threshold. As shown in Figure 23a, we can see that SAE + DEC has the best performance in the nanofiber dataset, while the performance of IAE + DEC is not as good as IAE. Figure 23b shows the ROC analysis for the carpet dataset in which both SAE + DEC and IAE + DEC have nearly 100% accuracy, showing that the DEC loss function has a positive effect on classification. The ROC analysis on the four models for the grid dataset is shown in Figure 23c, and their accuracies are all equal to or close to 100%. The main reason is that the grid image is regular and has an obvious defect pattern, so it is less likely to be misjudged during classification.

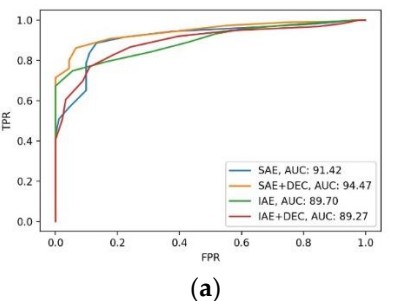
(a)

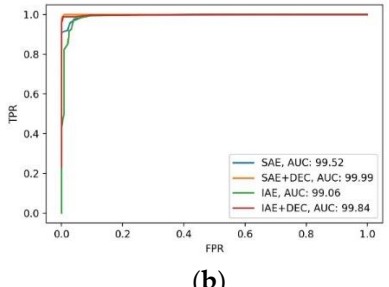
(b)

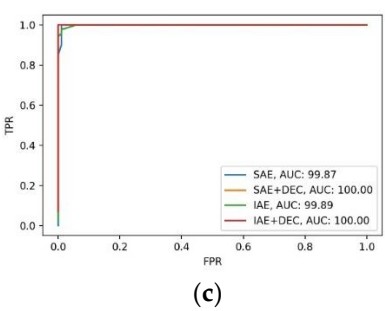
(c)

**Figure 23.** The ROC curves of patch image simulation for (**a**) nanofiber, (**b**) carpet, and (**c**) grid datasets.

In Tables 11–13, we analyze the classification accuracy for complete images that are spliced from the patch images back to complete images. The classification method is similar to the patch image classification. If a complete image includes a patch image classified as a defect image, the complete image is a defect image. Here, we ignore the analysis of classifying normal images for the nanofiber dataset because there is only one normal image in the dataset. It is obvious that integrating the DEC loss function can make the classification of normal images maintain a high classification accuracy, while the classification accuracy of the defective image is greatly improved as a whole.

**Table 11.** Classification accuracy for complete nanofiber images.

|  | Nanofiber | |
|---|---|---|
|  | **Good** | **Defect** |
| SAE | - | 95% |
| SAE + DEC | - | 100% |
| IAE | - | 100% |
| IAE + DEC | - | 100% |

**Table 12.** Classification accuracy for complete carpet images.

|  | Carpet | | | | | |
|---|---|---|---|---|---|---|
|  | **Good** | **Color** | **Cut** | **Hole** | **Metal** | **Thread** |
| SAE | 100% | 42.10% | 100% | 94.11% | 100% | 89.47% |
| SAE + DEC | 89.29% | 94.73% | 100% | 100% | 100% | 100% |
| IAE | 96.50% | 42.10% | 100% | 100% | 100% | 78.94% |
| IAE + DEC | 96.50% | 94.73% | 100% | 100% | 100% | 100% |

**Table 13.** Classification accuracy for complete grid images.

| | Grid | | | | | |
|---|---|---|---|---|---|---|
| | **Good** | **Bent** | **Broken** | **Glue** | **Metal** | **Thread** |
| SAE | 95.30% | 100% | 100% | 90.90% | 100% | 100% |
| SAE + DEC | 100% | 100% | 100% | 100% | 100% | 100% |
| IAE | 100% | 100% | 100% | 72.72% | 100% | 100% |
| IAE + DEC | 100% | 100% | 100% | 90.90% | 100% | 100% |

Figure 24 shows the ROC curves of the four models on the complete image of the carpet and grid datasets. This experiment also uses different classification thresholds to compare with the fourth Otsu threshold. In Figure 24a, the ROC curves show that classification accuracy of integrating DEC loss function is higher than others without integrating the DEC loss function. In Figure 24b, the AUC values of the four models in the grid image test are all greater than 99% because the texture pattern is regular.

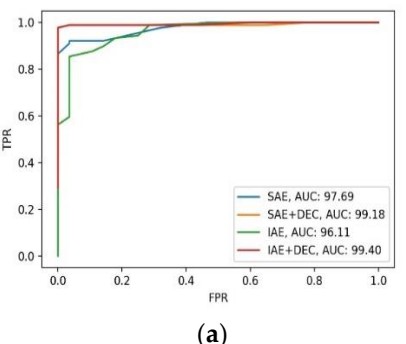
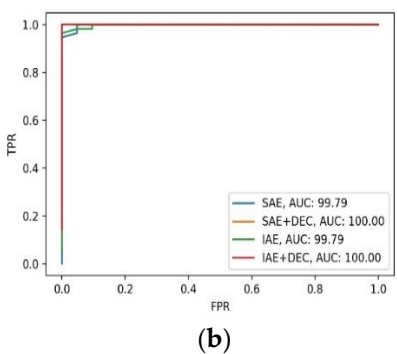

(**a**)                (**b**)

**Figure 24.** The ROC curves of complete image simulation for (**a**) carpet and (**b**) grid datasets.

Using the multi-thresholding method to determine whether the image is a defect image can avoid misjudgments caused by weak defective areas. For example, Figures 25c and 26c show that there are quite large differences in the grayscale distribution on the difference images shown in Figures 25a and 26a, but the classification results are correct, shown in Figures 25b and 26b. The reason is that the multi-thresholding method calculates the fourth threshold according to the image difference distribution. Therefore, even if the image difference is not obvious, the defect area can be detected. It is difficult to obtain the correct detection result by manually setting the threshold.

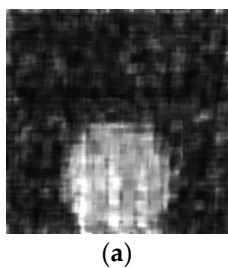
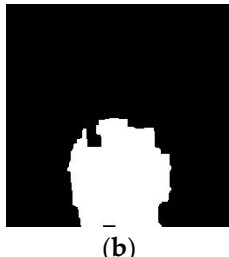
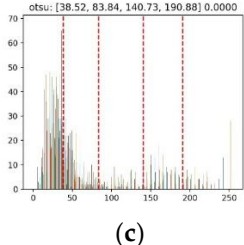

(**a**)                (**b**)                (**c**)

**Figure 25.** Analysis of using multi-thresholding to identify the obvious defect carpet image. (**a**) Obvious difference image. (**b**) Apply Otsu multi-thresholding method to detect the defective area. (**c**) The result of multi-thresholding.

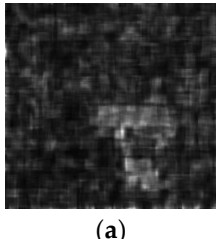 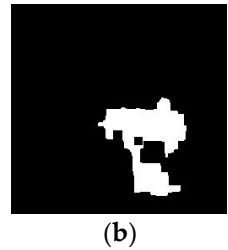 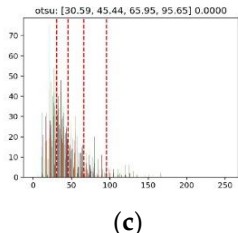

(**a**)　　　　　　　(**b**)　　　　　　　(**c**)

**Figure 26.** Analysis of using multi-thresholding to identify the weak defect carpet image. (**a**) Weak difference image. (**b**) Apply Otsu multi-thresholding method to detect the defective area. (**c**) The result of multi-thresholding.

### 4.4. Subjective Evaluation of Defective Area Detection

In addition to analyzing the classification accuracy, we also provide the subjective evaluation for the four kinds of Autoencoder models. In Figure 27, we compare the detection results of the four models on nanofiber, carpet, and grid datasets. Due to the assistance of the DEC loss function, the IAE + DEC and SAE + DEC models can have good completeness of repairing defect areas. At the same time, by observing the output image, it is found that the IAE + DEC model has the best repair capability, and the detection of defective areas is also the best in the subjective evaluation.

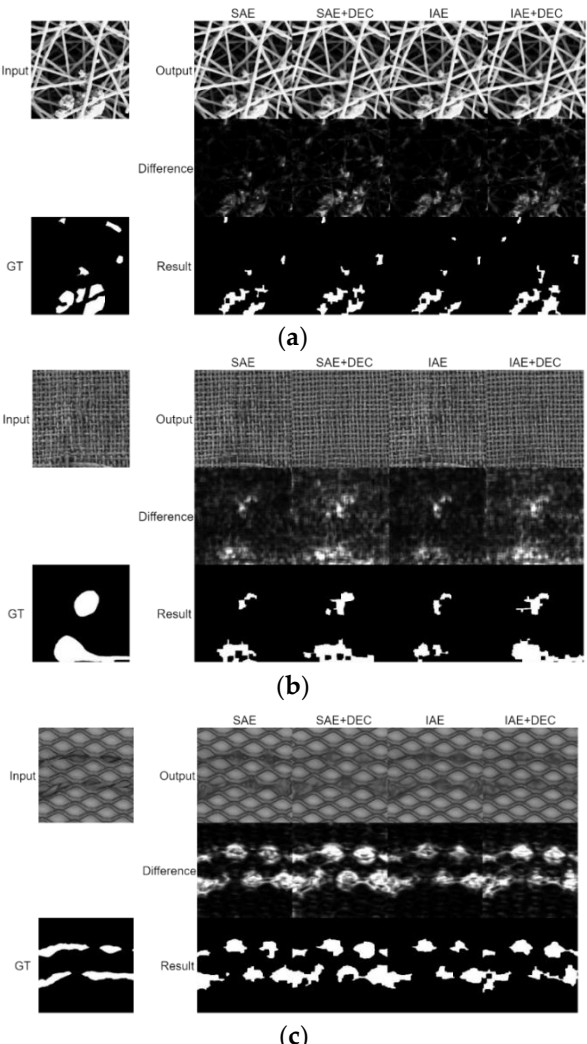

**Figure 27.** The defect detection results of the four models on (**a**) nanofiber, (**b**) carpet, and (**c**) grid datasets for the subjective evaluation.

### 4.5. Comparison with Methods in MVTec

Since there are no experimental data from other literature about the classification accuracy on the nanofiber dataset, we only compare the classification accuracy of carpet and grid datasets with other methods. The classification accuracy and the accuracy of the detected defective area are analyzed by using the complete images. In this paper, we compare our proposed method with the classification method in MVTec [14]. First, MVTec [14] selects some normal images as the training set and defines the minimum number of pixels for a defective area, that is, if the defect area is less than this number of pixels, it is judged as noise. Second, to determine the classification threshold, the threshold value is increased gradually from 0 to 255. When the maximum defect area in the training images is just below the defined minimum defective area, this threshold is used as the image classification threshold.

In Table 14, we compare the four proposed models with the models in [14]. In general, our proposed models outperform the models in [14] on both classifications of normal and defect images. Even if the DEC loss function is not included in the training model, the classification accuracy of defect images is higher than 80%. Among the four models, IAE + DEC has overall better performance. The main reason for improving the accuracy of image classification is that the multi-thresholding image segmentation algorithms provide adaptive thresholding for each test image to distinguish between normal and defective images. The method used in [14] sets a fixed classification threshold for each image, so it is difficult to have high classification accuracy.

**Table 14.** Comparison of the four proposed models with the models in [14].

| Model | Carpet | | Grid | |
| --- | --- | --- | --- | --- |
| | Good | Defect | Good | Defect |
| SAE | 100% | 85% | 95% | 98% |
| SAE + DEC | 89% | 98% | 100% | 100% |
| IAE | 96% | 84% | 100% | 94% |
| IAE + DEC | 96% | 98% | 100% | 98% |
| AE(SSIM) [14] | 43% | 90% | 38% | 100% |
| AE(L2) [14] | 57% | 42% | 57% | 98% |
| AnoGAN [14] | 82% | 16% | 90% | 12% |
| CNN Dict [14] | 89% | 36% | 57% | 33% |

In Table 15, we analyze the accuracy of the detected defective area with the methods in [14]. The method of analyzing the accuracy of the detected defective area is based on the method in [14] shown in Figure 28. The green area indicates the defective area that is predicted correctly, and the red area is the defective area that is missed in the prediction. The accuracy of the detected defective area is calculated as the ratio between the correct predicted area and the union of ground truth and predicted area.

**Table 15.** Analysis of the accuracy of detected defective area with the methods in [14].

| Models | Carpet | Grid |
| --- | --- | --- |
| SAE | 56% | 58% |
| SAE + DEC | 76% | 74% |
| IAE | 48% | 53% |
| IAE + DEC | 78% | 73% |
| AE(SSIM) [14] | 69% | 88% |
| AE(L2) [14] | 38% | 83% |
| AnoGAN [14] | 34% | 4% |
| CNN Dict [14] | 20% | 2% |

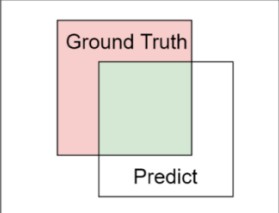

**Figure 28.** The method of analyzing the accuracy of detected defective area.

In Table 15, when the DEC loss function is not included, the highest accuracy of the detected defective area among the proposed models has a detection accuracy of only 50%. However, after the DEC loss function is included, the accuracy is increased to more than 70%, indicating that the DEC loss function improves the accuracy of the detected defective area. Among the four methods used in [14], the AE (SSIM) method is the best. The accuracy of the detected defective area for the grid dataset is 88% that is higher than our proposed models. However, from the classification accuracy for normal images 38% and defective images 100%, it can be seen that this model focuses on detecting defective images, but the misjudgment ratio of normal images is too high.

*4.6. Discussion*

This study aims to construct a novel Autoencoder that can have both normal area reconstruction and defective-area repair capabilities. Hence, a new model training method with DEC and SSIM loss functions is proposed. The ablation studies for MSE and SSIM loss functions and weighting analysis between SSIM and DEC loss functions are provided in Sections 4.1 and 4.2. The simulation results show that using SSIM is better for texture repair in the defective area for nanofiber images. It is also obvious that the defective area can be clearly found using SSIM, but the defective area cannot be found using MSE. Furthermore, in Table 6, the weighting ratio between SSIM and DEC loss functions for the Simple and Inception Autoencoders is important for both normal area reconstruction and defect area repair capabilities. However, on more complex nanofiber images, the classification accuracy of normal images is reduced because the DEC loss function can reduce the reconstruction capability on normal areas, as listed in Table 8. It is interesting that SAE + DEC outperforms IAE + DEC on complex nanofiber images, and this result still needs to be further explored and discussed.

In the following, the improvement of classifying defect and normal images is addressed by using the multi-thresholding method. The accuracy analysis for classifying normal and defect test images, subjective evaluation of defective area detection, and the comparison between our proposed method and the SOTA methods in [14] are conducted to verify the performance of the proposed method from Sections 4.3–4.5. The experimental results in Table 14 show that our proposed methods outperform the methods in [14] on both classifications of normal and defect images. The main reason for improving the accuracy of image classification is that our proposed Autoencoder can have both normal area reconstruction and defect area repair capabilities and the multi-thresholding image segmentation algorithm provides adaptive thresholding for each test image to distinguish between normal and defective images.

**5. Conclusions**

There are three main contributions in this study. First, we construct four kinds of Autoencoder models and combine the DEC clustering technology to strengthen the discrimination of the latent vectors. Second, an effective two-stage model training method is proposed to train the model with high normal area reconstruction and defect area repair capability. Finally, a novel defect image classification method between normal and defective images is proposed by using the multi-thresholding image-segmentation algorithm. The experimental results show that IAE + DEC has a better overall performance. Furthermore,

the accuracy of image classification is greatly improved by using the multi-thresholding image segmentation algorithms. The image classification accuracy for normal and defect images can be higher than 96%. However, the accuracy of classifying normal and defect patch images for the nanofiber dataset is lower than the regular patterns of grid and carpet images. This means the normal area reconstruction and defect area repair capability for complex texture patterns can be improved further in the future. For example, the self-attention mechanism can be integrated into the Autoencoder model to have higher defect-inspection accuracy.

**Author Contributions:** Conceptualization, C.-C.L.; methodology, C.-C.L. and Y.-D.C.; software, Y.-D.C.; validation, C.-C.L. and Y.-D.C.; formal analysis, C.-C.L.; investigation, C.-C.L.; resources, C.-C.L.; data curation, Y.-D.C.; writing—original draft preparation, C.-C.L. and Y.-D.C.; writing—review and editing, C.-C.L.; visualization, Y.-D.C.; supervision, C.-C.L.; All authors have read and agreed to the published version of the manuscript.

**Funding:** This research received no external funding.

**Institutional Review Board Statement:** Not applicable for studies not involving humans or animals.

**Informed Consent Statement:** Not applicable for studies not involving humans.

**Conflicts of Interest:** The authors declare no conflict of interest.

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
