# Peer review of "A Defect-Inspection System Constructed by Applying Autoencoder with Clustered Latent Vectors and Multi-Thresholding Classification"

_applsci, doi:10.3390/app12041883_

Round 1

Reviewer 1 Report

Decision: Major Revision

The idea presented in this paper is worthy and can be proved beneficial for the construction of a defect inspection system with an unsupervised learning model. The presentation of the paper is to convince but there are some major and minor comments and suggestions for improvement of this paper which should be addressed for possible acceptance in this journal. I looking forward to seeing these changes in the revised version.

Main points:

  1. Please explain why to use these datasets in the experimental design and mention the full form of datasets and define the DEC (before mentions and check the overall manuscript regarding this issue) in the abstract.
  2. As you mentioned you have used a novel autoencoder but I didn’t find any novelty in the corresponding section so it’s an existence technique. What makes the proposed method New and suitable for this unique task? What new development to the proposed method have the authors added (compared to the existing approaches)? These points should be clarified.
  3. The complexity of the proposed model and the model parameter uncertainty is not mentioned in the methodology section. It should be mentioned and experimentally proven to easily show the model significance.
  4. The current challenges are not crystal clearly mentioned in the introduction section.
  5. The authors need to proper attention to the experimental section. The performance of the proposed method should be better analyzed, commented and visualized in the experimental section.
  6. Definitions of accuracy are not clear and well defined have to be clear and distinct.
  7. The “Discussion” section should be added in a more highlighting, argumentative way. The author should analyze the reason why the tested results are achieved.
  8. More recently-published papers in the field of deep learning should be discussed in the Introduction/literature. The authors may be benefited by reviewing more papers such as DOI: 10.1109/ACCESS.2021.3093053. And 10.1016/j.future.2021.06.045.
  9. The readability and presentation of the study should be further improved. The paper suffers from language problems.
  10. Why is the proposed approach suitable to be used to solve the critical problem? We need a more convincing response to indicate clearly the SOTA development and how the penalty map differentiates the various edges as you mention in your method as a core part.
  11. Section Conclusion - Authors are suggested to include in conclusion section the real actual results for the best performance of their proposed methods in comparison towards other methods to highlight and justify the advantages of their proposed methods and future direction is not mentioned it should be added in detail in the revised version at the end of the conclusion section.

Reviewer 2 Report

The authors propose an unsupervised learning detection approach for surface defects. The paper is well organized. Some suggestions:
1. The multi-thresholding image segmentation algorithm has been applied in many references. What are the main innovations of the authors?  
2. I encourage the authors to open a demo code so that readers can verify it repeatably. 
3. The efficiency of the method is not analyzed, such as running time and speed.
4. I encourage the authors to add some case studies of failure for the method.
5. Actually, there are many publicly surface defect datasets. Why did the authors select only one dataset for validation? Theoretically speaking, more datasets can better verify the generalization of the proposed method.

Reviewer 3 Report

  • revise abbreviations in abstract
  • Good statement in Lines 45-50 as crucial issue of inspection systems for defect detection
  • There is used K-means clustering and preset is 40 clusters. Why exactly 40 clusters? What is impact of preset clusters number to the system? 
  • In comparison of SSIM and MSE methods in reconstruction error. It look very interesting, no only between methods but between the data of the fiber, carpet and grid. What are specifics between data and their impact to the system?
  • The described work is good. Do authors thinking about the possibilities of demonstrated system in case of different data, which contain:
    • Notations and text
    • Symbols
    • Patterns such as company logo etc.?
  • What is continuing this work? Are there new task or applications of described system to the future?

In general, described work is good. 

Round 2

Reviewer 1 Report

Comments: Need more attention and address all points with comprehensive and professional.

  1. I completely disagree with the author's responses. The authors didn’t respond to the questions and comments in a comprehensive manner as well as the paper still has many flows. The authors need to seriously follow the rules and address all questions and comments. (take it professionally)
  2. I have a question about the practical implementation and the computational novelty of this system (code should be provided or mention the like of code for verification). Authors need to convince readers about the applications of their system throughout the paper.
  3. In the response to the previous comment, lack of comparative analysis (compare with the wrong paper and with one). The experimental part is relatively weak. Firstly, the author chose to compare with more than two deep learning models, which have used the same data and solved a similar problem. So the performance improvement is unconvincing due to comparing with the wrong paper. Secondly, the analysis of the experimental results is not sufficient, and the results of Tables 14 and 15 are not reasonably explained.
  1. In the response to the previous comment, how you can convince people about your model without parameter setting. There are many models to provide good results than yours. I strongly disagree with your responses. (need explain parameters setting and prove with experiment)
  2. What you have done to reduce the model overfitting I didn’t find any info about this in the revised manuscript. It should be explained if your model is not overfitted give the reasons. According to figure 23 b, and c (ROC) your model is completely overfitted but I didn’t find any technique in the paper to solve this problem. (the results is vague)
  3. Disagree with your response 5. The complexity of the proposed model and the model parameter uncertainty is not mentioned in the methodology section. It should be mentioned and experimentally proven to easily show the model's significance. (this is not a response to tell us “we will consider in future work” Please consider in this one as well as in future)
  4. Your manuscript needs of ablation study because there is much confusion. I strongly recommend adding a dedicated ablation study table to prove your work.
  1. Disagree with your response 7. Don’t worry about the length of the paper and add a “Discussion” section, which should be shown more highlighting, argumentative way. The author should analyze the reason why the tested results are achieved.
  1. I completely disagree with your many responses. I hope you will re-considered it in this round.

Reviewer 2 Report

The manuscript quality improved and my quiestions were answered. I have no further comments. 

Author Response

Thank you for your positive comments.

Reviewer 3 Report

Thanks for revised manuscript.

Author Response

Thank you for your positive comments.